# Artificial Liver Support with CytoSorb and MARS in Liver Failure: A Retrospective Propensity Matched Analysis

**DOI:** 10.3390/jcm12062258

**Published:** 2023-03-14

**Authors:** Mihai Popescu, Corina David, Alexandra Marcu, Mihaela Roxana Olita, Mariana Mihaila, Dana Tomescu

**Affiliations:** 1Department of Anaesthesia and Intensive Care, “Carol Davila” University of Medicine and Pharmacy, 022328 Bucharest, Romania; 2Department of Anaesthesia and Intensive Care, Fundeni Clinical Institute, 022328 Bucharest, Romania; 3Department of Internal Medicine, Fundeni Clinical Institute, 022328 Bucharest, Romania

**Keywords:** acute liver failure, acute-on-chronic liver failure, liver support therapy, artificial liver support, molecular adsorbent recirculating system, CytoSorb

## Abstract

Background: Liver failure represents a life-threatening organ dysfunction with liver transplantation as the only proven curable therapy to date. Liver assist devices have been extensively researched to either bridge such patients to transplantation or promote spontaneous recovery. The aim of our study was to compare two such devices, the Molecular Adsorbent Recirculating System (MARS) and CytoSorb, in patients with liver failure. Methods: We retrospectively included 15 patients who underwent MARS during their intensive care unit stay and matched them to 15 patients who underwent hemoadsorption using CytoSorb. Clinical and paraclinical data obtained after each individual session, after the course of treatment, as well as at the end of the intensive care unit stay were compared between the two groups. Results: Single sessions of CytoSorb and MARS were both associated with a significant decrease in bilirubin (*p* = 0.04 and *p* = 0.04, respectively) and ammonia levels (*p* = 0.04 and *p* = 0.04, respectively), but only CytoSorb therapy was associated with a decrease in lactate dehydrogenase levels (*p* = 0.04) and in platelet count (*p* = 0.04). After the course of treatment, only CytoSorb was associated with a significant decrease in lactate (*p* = 0.01), bilirubin (*p* = 0.01), ammonia (*p* = 0.02), and lactate dehydrogenase levels (*p* = 0.01), while patients treated with MARS did not show any improvement in paraclinical liver tests. In addition, only CytoSorb treatment was associated with a significant improvement in the Model for End-Stage Liver Disease Score (*p* = 0.04). Conclusion: In conclusion, our results show a potential benefit of CytoSorb in rebalancing liver functional tests in patients with liver failure compared to MARS but the exact effects on patient outcome, including hospital length of stay and survival, should be further investigated in randomized control trials.

## 1. Introduction

Acute liver failure (ALF) is a potentially life-threatening condition that can lead to multiorgan failure [1,2] with mortality rates of 25–50% [3,4,5]. Acute-on chronic liver failure (ACLF) is a clinical syndrome defined by the acute deterioration of chronic liver disease and the rapid development of organ failures, associated with high short-term mortality [6]. The rapid deterioration and the development of organ dysfunctions is due to exogenous and endogenous precipitating factors. In most cases, bacterial infection or necrosis frees substances in the body, triggering an excessive inflammatory response. These factors are called pathogen- and damage-associated molecular patterns [7,8]. Most patients developing ACLF have pre-existing cirrhosis, which in itself is a hyperinflammatory state [9,10]. Another aggravating factor is the immune paralysis described by several studies [11,12,13,14,15], that prevents effective countermeasures against infection and makes patients prone to serious complications.

Several therapies have been tested for the replacement of hepatic functions. So far, liver transplantation (LT) is the only curative therapy available. Survival rates are good, but availability and eligibility for transplant differs by country [16]. In the CANONIC study, only 4.5% of ACLF patients received LT. Reportedly, low transplant rates are due to the high prevalence of infection and organ failure that leads to a “too sick to transplant” decision [17], thus waiting-list mortality exceeds 50% in this population [18]. As a bridging-to-transplantation or bridging-to-recovery, extracorporeal liver support devices can be used to aid the liver’s detoxification function by removing albumin-bound toxins and water-soluble substances [19].

Of these, MARS (Molecular Adsorbent Recirculating System) is one of the most used therapies with a long history in the treatment of patients with ACLF and ALF. In two recent meta-analyses, MARS therapy was the best option from the available treatments in reducing mortality but the data failed to demonstrate any significant improvements in-hospital mortality or severity of hepatic encephalopathy [20,21]. CytoSorb (CytoSorbents, Monmouth Junction, NJ, USA) has recently been introduced in clinical practice as an extracorporeal blood purification device aimed at controlling systemic inflammation. As most cytokines fall within the range of absorption (up to approximately 55 kDa), they may be rapidly eliminated from the bloodstream and, hence, rebalance severe systemic inflammation. Therefore, CytoSorb is primarily used in septic shock and other hyperinflammatory conditions to prevent the harmful effects of the cytokine and overwhelming inflammatory response [22]. According to the last analysis of the International CytoSorb Registry, ALF was the third most frequent indication for its use, and the treatment resulted in substantial removal of bilirubin [23]. Clinical data are lacking, but a recent in vitro study showed that compared to MARS, CytoSorb hemoperfusion more effectively leads to the faster removal of TNF-α, IL-6, and reduced indirect bilirubin and bile acid levels [24].

Based on this background, we decided to perform a retrospective comparison between MARS- and CytoSorb-treated patients. The aim of this study was to assess and compare the effects of the two liver support therapies on liver function and associated organ dysfunctions in critically ill patients with liver failure.

## 2. Materials and Methods

The ethical approval for the present study was provided by the Ethical Committee of Fundeni Clinical Institute, Bucharest, Romania (approval number 54229/2020). Due to the retrospective nature of the study, patient informed consent was waived.

Patient selection. All consecutive patients who underwent MARS therapy between January 2013 and December 2019 were included in the study—MARS group. The MARS group was matched based on the following variables: age, severity score (Acute Physiology and Chronic Health Evaluation II—APACHE II, Sequential Organ Failure Assessment Score—SOFA, Model for End-Stage Liver Disease Score—MELD), serum bilirubin, lactate levels, International Normalized Ratio (INR), values and dose of vasopressor support at time of the first extracorporeal liver support therapy, grade IV hepatic encephalopathy (HE), and need for mechanical ventilation (MV) with patients who underwent continuous renal replacement therapy (CRRT) in combination with hemoadsorption during the same time period—CytoSorb group.

Inclusion criteria were age over 18 years, proven ALF or ACLF, and need for either CytoSorb or MARS treatment. Exclusion criteria were death during treatment due to reasons other than the ALF (for example acute myocardial infarction, technical issues, etc.), indications for extracorporeal liver support therapy other than ALF or ACLF (for example toxin removal), and duration of extracorporeal liver support therapy for less than 4 h. For the present study, ALF was defined in accordance with the European Association for the Study of the Liver Guidelines [25] and ACLF was defined in accordance with the European Association for the Study of the Liver Guidelines [26].

Extracorporeal liver support therapy. All patients were treated per institutional protocol agreed upon by the intensive care unit (ICU) staff (standard medical care) and extracorporeal liver support therapy was applied in additional to standard medical care. In our institution, standard medical care is in accordance with the European Association for the Study of the Liver Guidelines [25]. The type of artificial liver support, either MARS or CytoSorb, as well as the time when the therapy was started was decided by the attending treating physician. Indications to start extracorporeal liver support were acute liver failure, at least grade II ACLF with either severe hepatic encephalopathy, acute kidney failure or severe hyperbilirubinemia (bilirubin levels > 30 mg/dL). The decision to start the therapy was made within 24 h after ICU admission. No blood products, including cryoprecipitate and fresh–frozen plasma, as well as prothrombin complex concentrate or fibrinogen concentrate, were administered during the therapy.

An extracorporeal liver support session was considered a single therapy with either MARS or CytoSorb. The time of each session was that of the maximum time for either MARS or CytoSorb recommended by the produce. An extracorporeal liver support treatment was defined as the total consecutive or non-consecutive sessions of extracorporeal liver support sessions performed in a single patient.

MARS therapy. MARS therapy was applied using a standard continuous veno-venous hemodiafiltration (CVVHDF) set (Prismaflex ST150 Set, Baxter Medical AB, Kista, Sweden) in conjunction with a MARS treatment kit (x-MARS, Baxter Medical AB, Kista, Sweden). The MARS therapy was performed using a Baxter Gambro MARS^®^ Molecular Adsorbent Recirculating System (Baxter Medical AB, Kista, Sweden) and the renal replacement therapy using a Prismaflex system (Baxter Medical AB, Kista, Sweden). The albumin system was primed with 500 mL of 20% human albumin and the renal replacement therapy circuit was primed with normal saline. Blood access was obtained using a double lumen hemodialysis catheter inserted in the femoral vein. A blood flow of 2–3 mL/kg/min was used in the blood circuit and a flow rate of 15% lower was applied in the albumin circuit. The CVVHDF settings were as follows: substitution of 10 mL/kg/h, dialysate of 15 mL/kg/h and ultrafiltration as deemed appropriate by the prescribing physician to control fluid balance. A PrismaSol BGK 4/2.5 (Baxter Medical AB, Kista, Sweden) solution was used as both a dialysate and replacement fluid. Anticoagulation was performed with unfractionated heparin in continuous infusion in the CRRT circuit to maintain an activated partial thromboplastin time 1.5–2 times above the upper reference level. A MARS session was performed for a maximum of eight hours.

CytoSorb therapy. The procedure was performed on Priasmaflex (Baxter Medical AB, Kista, Sweden) using a ST 150 hemofilter (Baxter Medical AB, Kista, Sweeden). Normal saline was used as a priming solution and PrismaSol BGK 4/2.5 (Baxter Medical AB, Kista, Sweden) solution was used for both substitution and dialysis. The CVVHDF prescription was the following: blood flow of 2–3 mL/kg/min, substitution of 10 mL/kg/h, dialysate of 15 mL/kg/h, and ultrafiltration as deemed appropriate by the prescribing physician to control fluid balance. Anticoagulation was performed with unfractionated heparin in continuous infusion in the CRRT circuit to maintain an activated partial thromboplastin time 1.5–2 times above the upper reference level. The CytoSorb cartridge (CytoSorbents Europe GMBH, Berlin, Germany) was placed upstream of the CVVHDF hemofilter. Each session of CVVHDF with hemoadsorption was performed for a maximum of 24 h.

Data collection. Data were collected from the available case records into a predefined case report form at T0 (baseline—demographic, laboratory and physiological parameters recorded at the time of ICU admission); then before and after each session of liver support therapy (T1: before, T1: after, Tn: before, and Tn: after, where n is the session treatment in a single patient). End of data collection (Te) is defined by the last laboratory and other data recorded at the end of ICU stay. Demographic data collected at ICU admission included age, gender, severity scores, body mass index, comorbidities, and liver diagnosis.

The following laboratory data were collected at the previous mentioned time points: lactate, creatinine, urea (BUN), bilirubin (total, direct, and indirect), sodium, INR, albumin, aspartate aminotransferase (AST), alanine aminotransferase (ALT), gamma-glutamyl transferase (GGT), ammonia, lactate dehydrogenase (LDH), hemoglobin (Hb), white blood cell count (WBC), platelet count (PLT), C-reactive protein (CRP), procalcitonin (PCT), fibrinogen, serum bicarbonate, and arterial partial pressure of oxygen (PaO_2_).

Clinical data collected at the forementioned time points included mean arterial pressure (MAP), heart rate (HR), noradrenaline dose, noradrenaline daily dose, need for mechanical ventilation, fraction of inspired oxygen (FiO_2_), PaO_2_/FiO_2_ ratio, and Glasgow Coma Scale (GCS).

The following severity scores were calculated at all time points: MELD score, SOFA score, and SOFA sub-scores (cardiovascular, respiratory, coagulation, liver, renal, central nervous system). The APACHE II score was calculated only at the time of ICU admission.

To assess the effects of liver support systems on different organ functions, the number of days on vasoactive drugs, number of days on renal replacement therapy other than liver assist therapy, and number of days on mechanical ventilation were also noted. Outcome measurements included length of ICU and hospital stay, 28-day survival, and liver transplantation performed during the hospital stay.

Endpoints. The primary endpoint was the degree of change in serum bilirubin as compared to baseline, i.e., the difference between the value at the end of therapy and before the first session was performed. Secondary endpoints included hemodynamic stability as defined by noradrenaline dose at the end of liver support therapy compared to baseline, effects of liver support therapies on the duration of other vital organ supports, on laboratory parameters, and on length of ICU and hospital stay, as well as 28-day mortality.

Statistical analysis. In total, 15 patients underwent MARS therapy, and 78 patients underwent CytoSorb therapy during the study period. To identify a matched group of CytoSorb patients for the MARS group, a propensity score analysis was applied. Factors associated with an unfavorable outcome were identified in the univariate analysis and then entered in a multivariate stepwise regression to identify independent factors. Using this model, each patient was designated a score based on their propensity to have an unfavorable outcome. The propensity-matched CytoSorb group was created for each propensity score by randomly selecting from the MARS group a CytoSorb patient with an equal propensity score. After matching the 2 cohorts for age, severity score (APACHE II, SOFA, MELD), serum bilirubin and lactate levels, and dose of vasopressor at T0, the groups were compared. Data are presented as mean ± standard deviation of the mean and median (min, max), and otherwise as percentage. Data distribution was examined to insure the proper statistical examination. Categorical variables were analyzed with Chi-square test and quantitative data were analyzed with independent samples *t*-test. The Mann–Whitney test was used when the analyzed data did not follow a normal distribution. Comparison between groups was performed using the two-way ANOVA or Kruskal–Wallis rank sum. Survival was assessed using Kaplan–Meier analysis and the *p* value was derived using log-rank test. All *p* values are two-tailed and a *p* value of less than 0.05 was considered statistically significant. Statistical analyses were performed using SPSS 19.0 (SPSS Inc.^®^, Chicago, IL, USA).

## 3. Results

Thirty patients were included in the final data analysis. Of these, 12 patients were diagnosed with ACLF (4 patients in the CytoSorb group and 8 patients in the MARS group) and 18 patients were diagnosed with ALF (11 patients in the CytoSorb group and 7 patients in the MARS group). The mean age in the study group was 38 ± 14 years and the mean MELD score was 31 ± 5 points. Patients in the CytoSorb group underwent a median number of 3 (2,3) sessions, with a total number of 44 sessions, and patients in the MARS group underwent a median number of 2 sessions (1–4), with a total number of sessions 28 sessions. There was no significant difference between the two groups in terms of demographic data, biochemical data, related organ dysfunction and severity scores. The characteristics of patients at the time of study enrolment for each treatment group are presented in Table 1.

### 3.1. Analysis of Biochemical Parameters

#### 3.1.1. Single Sessions

The changes in paraclinical data and severity scores before and after single sessions are summarized in Figure 1 and Table 2.

The use of CytoSorb was associated with a significant decrease in bilirubin levels from a median value of 10.4 mg/dL to 7.4 mg/dL, ammonia levels from a median value of 57 µmol/L to 45 µmol/L and LDH levels from 334 IU/L to 279 IU/L (Figure 1). A significant decrease in platelet count was also recorded from a median value of 76 × 10^3^/µL to 54 × 10^3^/µL (Figure 1). The use of a MARS session was associated with a significant decrease in bilirubin levels from a median value of 14.3 mg/dL to 12.0 mg/dL, ammonia levels from a median value of 41 µmol/L to 32 µmol/L (Figure 1). There was no difference between the two groups in the removal of either bilirubin or ammonia, but only CytoSorb was associated with a decrease in platelet count.

Regarding the rest of paraclinical data before and after a single session of liver support therapy, there were no significant differences neither within, nor between the two groups (Table 2).

#### 3.1.2. Course of Treatment

The overall effects (before and after the full course of treatment) of the liver support therapies are depicted in Figure 2 and Table 3.

The CytoSorb treatment was associated with a significant decrease in lactate levels, bilirubin, ammonia, and LDH levels (Figure 2) and an increase in bicarbonate levels (Table 3). We also observed a significant decrease in platelet count (Figure 1) and hemoglobin levels (Table 3). In the MARS treatment group, no significant difference was observed in paraclinical tests, except for a decrease in fibrinogen levels. INR values decreased in the CytoSorb group and increased in the MARS group. Although this was not statistically significant, inter-group comparison showed a significant difference in the INR dynamics between the two groups. There were no significant differences in the rest of the investigated laboratory parameters (Table 3).

### 3.2. Severity Scores and Organ Dysfunction

#### 3.2.1. Single Sessions

CytoSorb therapy were associated with a non-significant change in SOFA scores (Figure 3), as well as SOFA sub-scores (Table 2). In the MARS group, we also observed a non-significant change in SOFA score (Figure 3) as well as SOFA sub-scores (Table 2). A significant decrease in MELD score was observed in the CytoSorb group while a non-significant increase in MELD score was observed in the MARS group (Figure 3).

The use of CytoSorb was associated with non-significant changes in hemodynamic parameters: heart rate, mean arterial pressure and vasopressor support and Glasgow Coma Scale. The same non-significant changes were observed in the MARS group (Table 2).

#### 3.2.2. Course of Treatment

The overall effect of CytoSorb treatment was not associated with a significant decrease in SOFA score (Figure 3), but a significant increase in SOFA coagulation and a decrease in SOFA liver scores were observed (Table 3). We also noted a significant decrease in MELD score (Figure 3). The overall effect of MARS treatment was not associated with significant changes in SOFA or MELD scores (Figure 3) neither in SOFA sub-scores (Table 3). No significant differences in hemodynamic parameters or Glasgow Coma Scale were observed with either the CytoSorb or MARS treatment as depicted in Table 3.

### 3.3. ICU Outcome

The median hospital length of stay was 13.5 (6, 72) days and the median ICU length of stay was 13 (6, 60) days with no difference between the CytoSorb and MARS groups (*p* = 0.87 and *p* = 0.11, respectively). The median duration of RRT was 3 (0, 20) days, that of mechanical ventilation was 7 (0, 35) days, and vasopressor support was required for a median of 3 (0, 14) days. No significant differences were observed between the CytoSorb and MARS groups in terms of duration of mechanical ventilation (*p* = 0.32) and vasopressor support (*p* = 0.51), but patients in the CytoSorb group required longer RRT (6 (0, 20) vs. 1 (0, 4) days, *p* < 0.01). Five patients underwent emergency liver transplantation, three in the CytoSorb group and two in the MARS group.

In the CytoSorb group, paraclinical analysis at ICU admission compared to ICU at discharge showed a significant decrease in ALT, LDH levels, hemoglobin concentration, and platelet count and an increase in CRP levels (Appendix A). In the MARS group, we observed a significant increase in lactate and creatinine levels, a decrease in ALT, AST, and GGT levels, as well as in hemoglobin concentration and platelet count (Appendix A). A significant difference in inter-group kinetics of lactate levels was also noted with values being significantly higher in the MARS group as compared to the CytoSorb group. Ammonia values did not change significantly in either group; however, there was a significant inter-group difference in favor of CytoSorb therapy regarding the decreasing trend in ammonia levels (Appendix A).

Dynamics of severity scores between ICU admission and discharge demonstrated that CytoSorb was associated with a non-significant increase in SOFA and MELD scores; however, a significant increase in SOFA cardiovascular, SOFA coagulation, and SOFA renal sub-scores was observed. In the MARS group, we observed a significant increase in SOFA score and in SOFA cardiovascular, SOFA coagulation, and SOFA renal sub-scores (Appendix A).

Twenty-eight-day survival was 50% (*n* = 15), with no difference between the CytoSorb and MARS group (53.3% vs. 46.7%, *p* = 0.72) (Figure 4).

## 4. Discussion

Our study compared two liver support systems in patients admitted to the ICU with liver failure. The data were collected and analyzed before and after each individual session, before and after the full course of liver support treatment, and at ICU admission and discharge. The two groups consisted of severely ill patients with liver failure as demonstrated by the high severity scores (SOFA and MELD score) as well as high levels of bilirubin at the time of therapy start. Overall, both therapies were associated with a paraclinical improvement in liver tests demonstrated by a significant decrease in bilirubin, ammonia, lactate, transaminase, and LDH levels. Some of these changes were more pronounced in the CytoSorb group compared to the MARS group, but whether this statistical significance can be translated into a better outcome must be determined in adequately powered studies.

The use of a single session of CytoSorb therapy was associated with a significant decrease in bilirubin, ammonia, and LDH levels. This was also noted by Dhokia et al. [27] who demonstrated in their case series a paraclinical improvement in liver functional tests with the use of CytoSorb. Additionally, in a single case report, Faltlhauser et al. [28] showed a significant decrease in serum transaminases in a patient with severe liver failure; however, to date, no large-scale studies have demonstrated a significant decrease in serum transaminases associated with the use of CytoSorb. Our data also failed to demonstrate a significant decrease in ALT and AST levels. This may be due to the fact the median transaminase levels in our patients were only moderately increased to twice the upper normal value and only a few patients in the CytoSorb group had a marked increase in ALT and AST values and hence, our results may not fully reflect the extent of serum transaminase removal. Inflammatory markers demonstrated only a mild, non-significant decrease. This is also due, on the one hand, to the low pre-treatment levels of inflammatory markers and, on the other hand, to the fact that we could only measure standard markers and no specific cytokine levels could be determined. Nevertheless, the current literature supports the efficacy of CytoSorb in removal of pro-inflammatory cytokines and rebalancing the immune system. In a case series of septic patients, Mehta et al. [24] demonstrated a significant reduction in inflammatory markers, especially interleukin-6, interleukin-10, and tumor necrosis factor-α. Furthermore, in a case report by Tomescu et al. [29], the authors observed a significant decrease in pro-inflammatory cytokines and a rebalancing of the immune response in a patient with primary graft non-function after liver transplantation. As severe systemic inflammation represents a key aspect of both acute liver failure [30] and acute-on-chronic liver failure [10], the potential benefits of rebalancing pro- and anti-inflammatory cytokines with the aid of CytoSorb may be associated with an improved outcome in these patients and needs further evaluation.

MARS also demonstrated a significant decrease in bilirubin and ammonia levels. This has also been demonstrated by Novelli et al. [31]. By contrast with their study, we did not observe a significant change in lactate and creatinine levels. This may be since the baseline levels were significantly lower in our group compared to those reported by Novelli. This also applies to the improvement in hepatic encephalopathy as assessed by the GCS score. As in the CytoSorb group, we did not observe a significant improvement in serum transaminases probably due to the median low baseline levels. This is in accordance with the study published by Donati et al. [32] who reached the same conclusions. Schmidt has demonstrated that MARS induces a significant improvement in hemodynamic parameters by rebalancing the immune response [33]. Unfortunately, we were not able to record the full range of parameters measured invasively but our study showed no improvement in mean arterial pressure. This may be due to the low-normal baseline values for arterial pressure. However, even in patients with normal blood pressure, an improvement in hemodynamic parameters have been demonstrated [34,35], and hence, the exact extent of hemodynamic improvement remains debatable. As previously mentioned, severe inflammation represents a major pathophysiological mechanism behind liver failure. However, the clearance of inflammatory markers by MARS has not been fully demonstrated by the current literature. Sen et al. [36], demonstrated an improvement in clinical and paraclinical parameters but failed to show a significant decrease in serum cytokines. This non-decrease in inflammatory markers was also demonstrated by other authors [37,38]. Unfortunately, we were unable to measure specific inflammatory cytokines, but we did not observe a significant change in white blood cells count and C-reactive protein levels.

We observed a drop in platelet count in both groups that reached statistical significance in the CytoSorb group. However, bleeding complications were not observed in either group. It is difficult to define whether the drop in platelet count was due to the devices per se or whether it was the result of the extracorporeal circulation itself. Alharthy et al. [39] also reported thrombocytopenia as a side effect of CytoSorb therapy in critically ill patients with COVID-19 and acute kidney injury, while Paul et al. [40] did not find a significant change in platelet count in septic patients. The underlying cause of thrombocytopenia in patients with severe liver disease is multifactorial: activation of platelets by the renal replacement therapy circuit and platelet aggregation within the CytoSorb filter or the CRRT filter, secondary to the use of heparin as an anticoagulant and heparin-induced thrombocytopenia, or a progression of liver failure associated coagulopathy [41]. As CRRT itself is associated with a decrease in platelet count [42], further research comparing CRRT to CRRT and CytoSorb is needed to assess whether the decrease in platelet count is more significant when CytoSorb is added.

Our results demonstrated a significant reduction in lactate, bilirubin, and ammonia levels after a CytoSorb treatment, while the use of a MARS treatment was not associated with the same decrease. Ocskay et al. [23] demonstrated a decrease in liver functional tests, while Tomescu et al. [43] demonstrated a significant decrease in ammonia, bilirubin, and lactate in patients treated with CytoSorb. Comparing our results with the existing literature, as well as different studies published on liver support therapies may be biased as there is no universally applied protocol for the use of such devices in patients with liver disease regarding the number of sessions applied and the time lag between individual sessions. Moreover, the CytoSorb treatment consisted of a median of three consecutive sessions, while the MARS treatment consisted of only a median of two consecutive sessions and, hence our results on the overall effect of artificial liver treatment may be biased by this difference and our results should be interpreted with caution. Until a large-scale randomized control trial is performed, the effects of a single session of liver support therapy may provide a more reliable picture on the dynamics of liver functional tests.

Both MARS and CytoSorb treatments were associated with a significant decrease in serum transaminases, and CytoSorb was also associated with a significant decrease in GGT and LDH levels. However, as previously mentioned, the median values at the start of the therapies were only slightly increased above the upper normal value and hence the decreasing trend needs further confirmation by larger cohorts of patients. Patients in the CytoSorb group experienced a significant decrease in hemoglobin levels, while patients in the MARS group did not. As this decrease was observed over a median period of more than 72 h, we cannot assess if it is due to erythrocyte entrapment or destruction within the CytoSorb/CRRT filter as demonstrated by Al-Dorzi et al. [44] or to critical illness and the frequent blood withdrawal in the ICU [45]. A decrease in platelet count was observed for both MARS and CytoSorb groups. The mechanisms underlining this effect have been previously described. Nevertheless, platelet count should be tightly monitored during liver support therapies to avoid severe, life-threatening thrombocytopenia and guide appropriate measures. However, the shorter duration of treatment in the MARS group may be associated with a slower decline in platelet count and hemoglobin levels and hence offer a better protection against hemorrhagic or anemic complications and a decreased need for blood products’ administration.

An increase in C-reactive protein was observed in both groups but it reached statistical significance only during CytoSorb therapy. This increase in CRP may be due to either a progression of systemic inflammation due to aggravated liver failure [8] or bacterial sepsis [46] or it may be a marker of liver recovery. Due to the potential central role of systemic inflammation in both acute and acute-on-chronic liver failure, future research is needed to fully characterize the role of liver support devices in rebalancing the inflammatory response.

Neither CytoSorb nor MARS was associated with a significant change in SOFA score. In the MARS group, none of the scores changed significantly after the therapy. This is in discordance with the study by Guo et al. [47] who demonstrated a significant decrease in SOFA score in patients with severe liver failure and multiple system organ failure and that by Steiner et al. [34] who demonstrated a decreased MELD score associated with the use of MARS. In the CytoSorb group, we observed a statistically significant change in SOFA liver sub-score due to the significant decrease in bilirubin levels that was counterbalanced by an increased SOFA coagulation due to the previously mentioned decrease in platelet count. The improvement in liver functional tests in the CytoSorb group was also demonstrated by a 10-point decrease in the median MELD score, compared to a non-significant increase in the MARS group. However, both SOFA and MELD score only consider a limited number of variables and their sensitivity for early mortality and morbidity is suboptimal and hence, more specific and sensitive scores should be developed for critically ill patients with liver failure [48].

As most hemodynamic parameters were within the normal range at patients’ inclusion, no significant changes in heart rate and blood pressure were noted, demonstrating that both treatments induce minimal changes in systemic hemodynamics. However, in a study published by Schmidt et al. [49], the authors demonstrated that MARS therapy was associated with an increase in mean arterial blood pressure, decrease in heart rate, and normalization of cardiac index. Additionally, in a case report published by Tomescu et al. [29], CytoSorb therapy was associated with improved hemodynamics due to a rebalanced systemic inflammation. Unfortunately, we were not able to measure specific cytokine levels. As severity of systemic inflammation strongly correlates with hemodynamic instability and mortality [50,51], applying liver support systems after hemodynamic instability may be too late to have a definitive impact on outcome, but this needs further validation in future studies.

The comparison of liver functional tests between ICU admission and discharge demonstrated a significant decrease in serum transaminases in both groups. In the MARS group, a significant increase in lactate levels was noted. This may be due to a progression of liver failure. However, the increase in lactate levels may be due to other extra-hepatic causes [52]. Nevertheless, high lactate levels have been associated with increased mortality in critically ill patients with liver failure [53]. Despite the higher levels of lactate in the MARS group compared with the CytoSorb group, no significant difference in mortality was observed.

There are some limitations of our study that need to be considered. First, it was a retrospective, single center study on a relatively small number of patients. Therefore, the potential benefits of CytoSorb therapy over MARS observed in our study should be tested in randomized control trials. Second, no universal criteria exist for both ICU admission and discharge of patients with liver failure, as well as the perfect timing for applying extracorporeal liver support therapies. Although we applied local protocols, this may not be identical between different liver centers, and this makes the comparison of our results with those reported by other studies subject to error. Third, as previously mentioned, the number of therapies applied in each patient was based on previously published studies from our center as well as the clinical decision by the attending physician since a consensus on this topic does not exist between centers. Fourth, there was a significant difference in the number of sessions applied between the two groups and this introduces a significant bias when comparing the overall effects of the two therapies. Regarding this, our results need to be confirmed by other studies in which patients received the same number of sessions. Finally, the rebalancing of the immune response is considered a key effect of CytoSorb. Unfortunately, we were unable to determine pre- and post-therapy levels for specific pro- and anti-inflammatory cytokines. The correlation between the dynamics of inflammatory markers and improvement in liver function needs further research.

## 5. Conclusions

In conclusion, the use of CytoSorb was associated with a paraclinical improvement in liver functional tests demonstrated by a significant decrease in bilirubin, ammonia, lactate, transaminase, and LDH levels that were more pronounced compared to the MARS group and thus CytoSorb may provide a more extensive biochemical control of liver failure compared to MARS. However, their impact on hospital length of stay and patient survival needs further evaluation in large-scale randomized control trials.

## Figures and Tables

**Figure 1 jcm-12-02258-f001:**
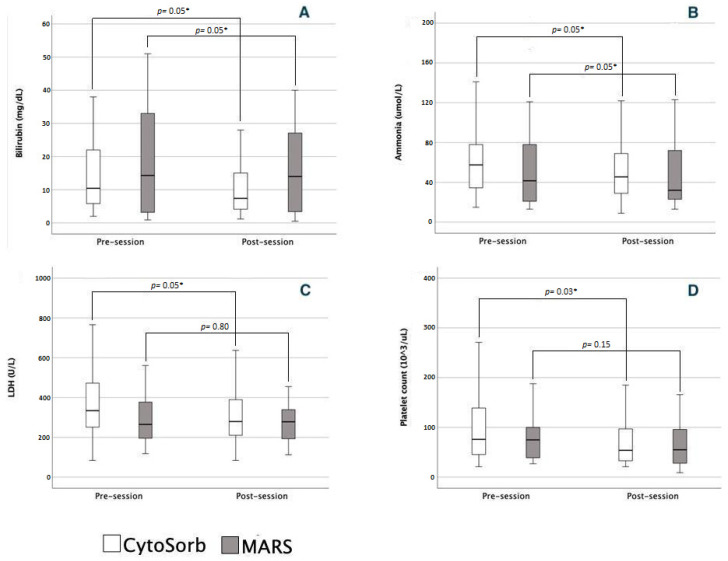
Comparison of biochemical parameters before and after a single session of liver support therapy. (**A**) Bilirubin; (**B**) ammonia; (**C**) lactate dehydrogenase; (**D**) platelet count. A session has been defined as a single session of extracorporeal liver support. * denotes a *p* value < 0.05.

**Figure 2 jcm-12-02258-f002:**
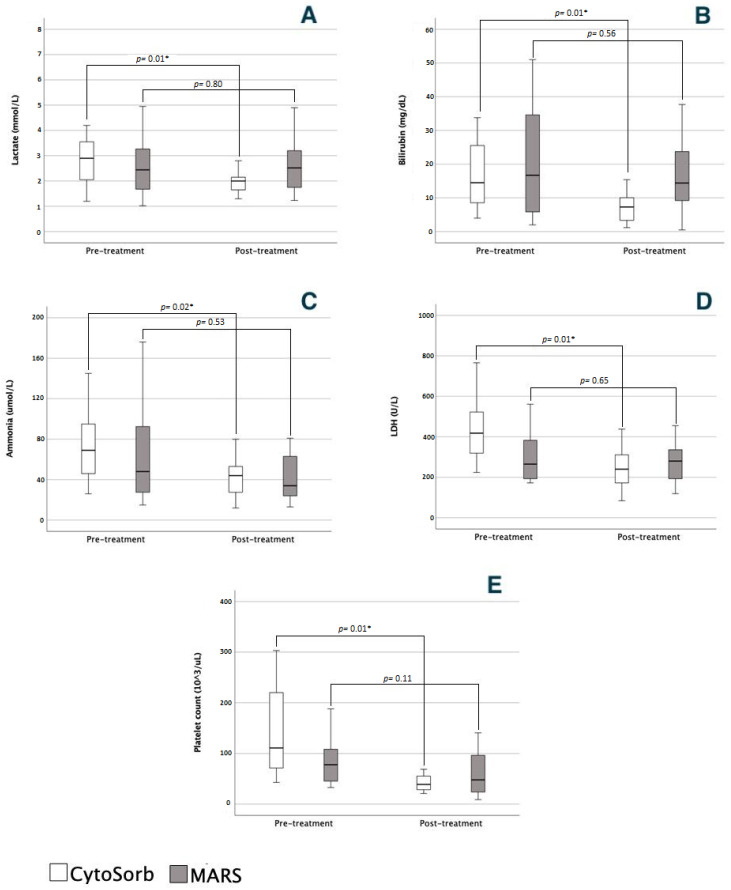
Comparison of biochemical parameters before and after a liver support treatment. (**A**) Lactate; (**B**) bilirubin; (**C**) ammonia; (**D**) lactate dehydrogenase; (**E**) platelet count. A liver support treatment is defined as the sum of all consecutive extracorporeal liver support sessions in a single patient. * denotes a *p* value < 0.05.

**Figure 3 jcm-12-02258-f003:**
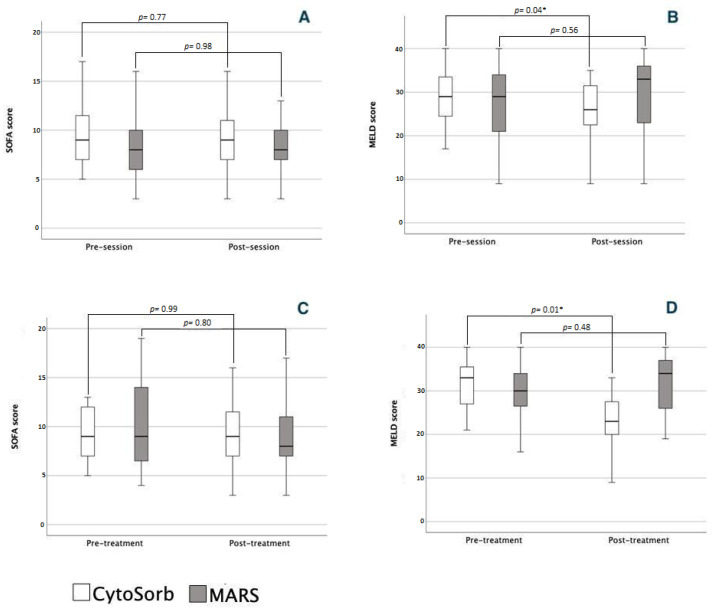
Comparison of severity scores before and after a single session of liver support therapy and before and after a liver support treatment. (**A**) Comparison of SOFA score after a single session; (**B**) comparison of MELD score after a single session; (**C**) comparison of SOFA score after a liver support treatment; (**D**) comparison of MELD score after a liver support treatment. A session is defined as a single session of extracorporeal liver support. A liver support treatment is defined as the sum of all consecutive extracorporeal liver support sessions in a single patient. * denotes a *p* value < 0.05.

**Figure 4 jcm-12-02258-f004:**
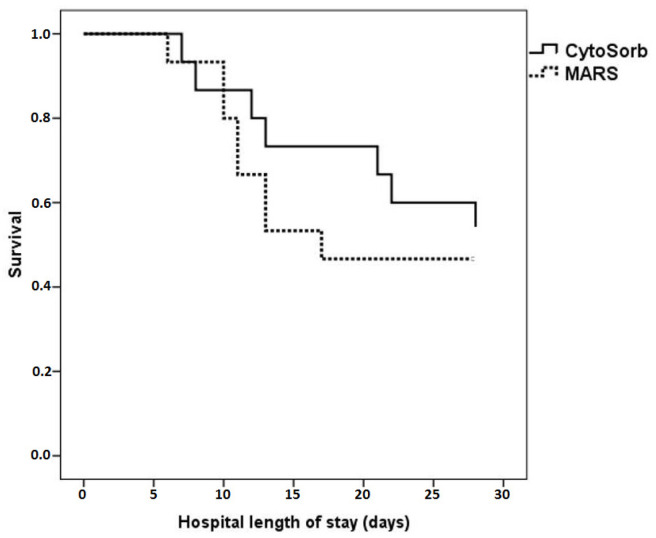
Twenty-eight-day outcome in patients undergoing liver support therapy.

**Table 1 jcm-12-02258-t001:** Baseline characteristics at patient inclusion.

Parameter	CytoSorb Group (*n* = 15)	MARS Group (*n* = 15)	*p* Value
Age (years)	37.4 ± 14.2	39.8 ± 14.8	0.72
MELD Score	31.9 ± 5.8	30.0 ± 5.4	0.31
SOFA Score	9.2 ± 2.8	8.2 ± 3.1	0.97
APACHE II Score	14.6 ± 7.4	14.5 ± 4.4	0.17
Bilirubin (mg/dL)	17.0 ± 9.7	20.8 ± 16.5	0.33
Albumin (mg/dL)	2.7 ± 0.5	2.9 ± 0.5	0.72
INR	3.2 ± 2.3	2.8 ± 1.1	0.32
Lactate (mmol/L)	2.8 ± 0.9	2.6 ± 1.3	0.30
Grade IV HE	33% (*n* = 5)	40% (*n* = 6)	0.79
Need for MV (%)	27% (*n* = 4)	40% (*n* = 6)	0.70

Legend. MELD, Model for End-Stage Liver Disease; SOFA, Sequential Organ Failure Assessment; APACHE II, Acute Physiology and Chronic Health Evaluation II; INR, International Normalized Ratio; HE, hepatic encephalopathy; MV, mechanical ventilation.

**Table 2 jcm-12-02258-t002:** Comparison on clinical and paraclinical data and severity scores before and after a single session of liver support therapy.

	CytoSorb Group			MARS Group			*p* Value between Groups
	Before Session	After Session	*p* Value	Before Session	After Session	*p* Value	
Lactate (µmol/L)	2.2 (0.1, 4.2)	2.1 (0.8, 3.7)	0.07	1.7 (0.8, 5.8)	2.1 (0.8, 19.0)	0.83	0.11
Creatinine (mg/dL)	0.8 (0.2, 6.0)	0.7 (0.2, 5.1)	0.38	0.5 (0.1, 3.3)	0.5 (0.2, 2.1)	0.47	0.98
BUN (mg/dL)	30 (3, 276)	25 (2, 201)	0.45	29 (11, 188)	29 (7, 136)	0.35	0.79
Sodium (mmol/L)	138 (125, 147)	139 (123, 150)	0.39	137 (121, 150)	136 (124, 145)	0.83	0.42
INR	2.6 (1.3, 11.2)	2.4 (1.2, 5.0)	0.46	2.3 (1.2, 5.4)	2.8 (1.2, 13.2)	0.24	0.02
Albumin (g/dL)	2.7 (1.7, 4.0)	2.8 (1.8, 4.0)	0.46	2.9 (1.17, 3.8)	2.7 (1.9, 3.8)	0.50	0.29
AST (U/L)	97 (24, 8219)	90 (21, 3942)	0.26	139 (35, 8219)	130 (38, 5207)	0.83	0.91
ALT (U/L)	142 (16, 8332)	91 (16, 3735)	0.38	114 (7, 7550)	103 (18, 6412)	0.71	0.93
GGT (U/L)	43 (17, 489)	38 (13, 361)	0.18	38 (16, 200)	35 (8, 157)	0.36	0.81
Hemoglobin (g/dL)	9.8 (6.7, 14.2)	9.7 (7.2, 14.1)	0.24	8.0 (6.9, 11.8)	7.8 (7.0, 11.7)	0.52	0.85
WBC (×10^3^/μL)	10.0 (2.9, 34.2)	9.9 (2.9, 25.3)	0.48	8.2 (1.6, 16.7)	9.1 (2.1, 27.1)	0.90	0.38
PCT (ng/mL)	0.8 (0.0, 5.8)	0.5 (0.0, 4.7)	0.22	0.6 (0.1, 2.5)	0.5 (0.2, 2.3)	0.95	0.47
CRP (mg/L)	7.0 (0.5, 137.0)	5.2 (0.5, 91.0)	0.59	14 (2, 75)	21 (1, 112)	0.83	0.53
Fibrinogen (mg/dL)	141 (60, 387)	142 (73, 232)	0.94	125 (60, 286)	108 (48, 123)	0.07	0.16
HCO_3_ (mmol/L)	23 (17, 27)	23 (21, 27)	0.13	24 (14, 32)	24 (11, 29)	0.47	0.73
SOFA CV	0 (0, 4)	0 (0, 4)	0.75	0 (0, 4)	0 (0, 4)	0.62	0.91
SOFA Resp	1 (0, 3)	1 (0, 3)	0.95	2 (0, 3)	1 (0, 3)	0.80	0.85
SOFA Coag	2 (0, 3)	2 (0, 3)	0.10	2 (0, 3)	2 (0, 4)	0.45	0.45
SOFA Liver	3 (2, 4)	3 (1, 4)	0.10	4 (0, 4)	4 (0, 4)	0.84	0.72
SOFA Renal	0 (0, 4)	0 (0, 4)	0.37	0 (0, 3)	0 (0, 3)	0.84	0.21
SOFA CNS	1 (0, 4)	1 (0, 4)	0.65	0 (0, 4)	0 (0, 4)	0.85	0.72
GCS (points)	12 (3, 15)	13 (3, 15)	0.59	15 (3, 15)	15 (2, 15)	0.98	0.09
MAP (mmHg)	75 (68, 97)	75 (68, 92)	0.94	83 (68, 104)	81 (61, 105)	0.77	0.32
HR (bpm)	83 (52, 125)	82 (52, 115)	0.34	87 (52, 109)	86 (56, 128)	0.74	0.64

Legend. Data are presented as median (min, max). BUN, urea; INR, International Normalized Ratio; AST, aspartate aminotransferase; ALT, alanine aminotransferase; GGT, gamma-glutamyl transferase; WBC, white blood cell count; CRP, C-reactive protein; PCT, procalcitonin, SOFA, Sequential Organ Failure Assessment; CV, cardiovascular; Resp, respiratory; Coag, coagulation; CNS, central nervous system; GCS, Glasgow Coma Scale; MAP, mean arterial pressure; HR, heart rate.

**Table 3 jcm-12-02258-t003:** Comparison on clinical and paraclinical data and severity scores before and after a liver support treatment.

	CytoSorb Group			MARS Group			*p* Value between Groups
	Before Treatment	After Treatment	*p* Value	Before Treatment	After Treatment	*p* Value	
Creatinine (mg/dL)	1.0 (0.2, 6.0)	0.7 (0.4, 2.1)	0.18	0.7 (0.2, 3.3)	0.6 (0.2, 1.8)	0.48	0.55
BUN (mg/dL)	60 (3, 276)	27 (2, 94)	0.11	29 (11, 188)	32 (7, 109)	0.50	0.15
Sodium (mmol/L)	138 (125, 141)	140 (126, 150)	0.21	137 (126, 150)	137 (124, 145)	0.95	0.18
INR	2.5 (1.5, 11.2)	2.1 (1.2, 3.4)	0.12	2.6 (1.4, 5.4)	2.8 (1.2, 13.2)	0.49	0.04
Albumin (g/dL)	2.7 (1.7, 3.6)	3.0 (2.0, 4.0)	0.10	3.1 (2.1, 3.7)	2.7 (1.9, 3.8)	0.53	0.08
AST (U/L)	98 (46, 8219)	56 (21, 3942)	0.05	133 (35, 8219)	123 (38, 2703)	0.59	0.85
ALT (U/L)	257 (31, 8332)	85 (20, 1043)	0.12	108 (7, 7550)	94 (18, 2639)	0.56	0.71
GGT (U/L)	58 (28, 489)	33 (13, 309)	0.02	38 (16, 200)	35 (8, 157)	0.43	0.31
Hemoglobin (g/dL)	10.6 (6.7, 14.2)	9.0 (7.2, 12.1)	0.03	8.6 (6.9, 11.4)	8.3 (7.0, 11.5)	0.32	0.01
WBC (×10^3^/μL)	12.3 (6.8, 34.2)	10.9 (2.9, 20.4)	0.22	9.0 (1.6, 13.7)	9.1 (2.1, 27.1)	0.91	0.14
PCT (ng/mL)	0.9 (0.1, 5.8)	0.4 (0.0, 4.0)	0.12	0.6 (0.1, 2.5)	0.5 (0.2, 2.2)	0.74	0.42
CRP (mg/L)	12.0 (0.5, 137.0)	5.3 (1.2, 90.0)	0.49	14.0 (2.1, 56.0)	23.0 (1.8, 112.0)	0.64	0.62
Fibrinogen (mg/dL)	145 (83, 387)	146 (74, 232)	0.74	125 (82, 286)	86 (48, 229)	0.01	0.38
HCO_3_ (mmol/L)	21 (17, 27)	23 (21, 27)	0.02	23 (14, 32)	24 (11, 29)	0.50	0.35
SOFA CV	0 (0, 0)	0 (0, 4)	0.90	0 (0, 4)	0 (0, 4)	0.99	0.38
SOFA Resp	1 (0, 3)	2 (0, 2)	0.93	2 (0, 3)	1 (0, 3)	0.30	0.57
SOFA Coag	1 (0, 3)	3 (1, 3)	0.01	2 (0, 3)	2 (1, 4)	0.38	0.12
SOFA Liver	4 (2, 4)	3 (1, 4)	0.02	4 (2, 4)	4 (0, 4)	0.96	0.13
SOFA Renal	0 (0, 4)	0 (0, 4)	0.23	0 (0, 3)	0 (0, 3)	0.80	0.56
SOFA CNS	1 (1, 4)	1 (0, 4)	0.74	1 (0, 4)	0 (0, 4)	0.80	0.93
GCS (points)	13 (3, 14)	14 (3, 15)	0.62	14 (3, 15)	15 (3, 15)	0.87	0.85
MAP (mmHg)	72 (68, 97)	75 (68, 87)	0.64	85 (70, 104)	81 (61, 85)	0.08	0.12
HR (bpm)	83 (65, 125)	82 (55, 95)	0.28	87 (67, 109)	86 (56, 124)	0.87	0.31

Legend. Data are presented as median (min, max). BUN, urea; INR, International Normalized Ratio; AST, aspartate aminotransferase; ALT, alanine aminotransferase; GGT, gamma-glutamyl transferase; WBC, white blood cell count; CRP, C-reactive protein; PCT, procalcitonin, SOFA, Sequential Organ Failure Assessment; CV, cardiovascular; Resp, respiratory; Coag, coagulation; CNS, central nervous system; GCS, Glasgow Coma Scale; MAP, mean arterial pressure; HR, heart rate.

## Data Availability

Data can be obtained from the corresponding author upon reasonable request.

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
