# Peer review of "Artificial Liver Support with CytoSorb and MARS in Liver Failure: A Retrospective Propensity Matched Analysis"

_jcm, 2023, doi:10.3390/jcm12062258_

Round 1

Reviewer 1 Report

A well written paper, some minor and one major comment for me:

Minor

Extracorporeal liver support therapy: in my opinion it must be better presented which type is the standard medical therapy.

In addition, the nature of the doctors personal decision in favour of liver support therapy should be better described. That means: when was the decision made and on what basis? Please describe better.

Baseline characteristics: Bilirubin levels in CS group were 17 and in MARS group 20.8 mg/dl. This implicates that CS group wasn`t as severe ill in behalf of the liver function? Please describe the role of that.

Major

The big difference in both therapys is, that a MARS session was 8 hrs and a CS session was 24 hrs. In my opinion this means that as described in the results the median treatment time was 16 hrs for MARS and 72 hrs for CS. This alone can lead to these different results. Please describe clearly.

Author Response

Dear reviewer, 

Thank you very much for your time and effort in reviewing our study. We have taken into account your observations and they have been included in the text and marked with trakchanges. We also offer a point by point response to them bellow. 

Answer to minor observations:

  1. indeed, standard medical therapy may vary between centres. In our institution the EASLD guidelines have long been implemented as standard medical therapy. We have added this observation in the methods section.  
  2. We have introduced our criteria, as stated in our local protocol, to start liver support therapy. The decision was made by the attending physician to on when to start the therapy within 24 hours based on the time it took to collect the clinical and lab test it needed. To answer your remark on how one therapy was chosen over another unfortunately this decision was made on the availability of kits and knowledge to use them (especially MARS) as some of the patients were admitted during the night or weekend. From our databased we made a propensity score in order to include in both groups patients that had no statistical difference in baseline characteristics. 
  3.  As you can see from the same tabel although the CS group had apparently lower levels for bilirubin, these were not statistically significant (p=0.33) and hence the groups were matched in this parameter (as in all others included in the propensity match).  

Answer to major observation: 

Thank you for this very interesting question and the possibility of share more light on it. There are many point to discuss and we will discuss them systematically: 

  1. in regard to the difference in time of extracorporeal liver support therapy. the maximal time recommended by each of the companies who produce the respective filters as this is considered to be the time at which the filter is no more efficient (8 h for MARS and 24 h for CS). Moreover, this also is the time that has been considered the standard time for each of the procedure in the most important studies published in current literature: MARS (https://doi.org/10.1371/journal. pone.0175529, DOI: 10.1097/SLA.0000000000002361) including, of course, in the largest trial on the issue of using MARS in ACLF - the RELIEF trail (doi:10.1002/hep.26185) and for CS (doi:10.1177/0391398820981383, https://doi.org/10.3390/jcm10215182). If we are allowed to make a comparison, when comparing drug A to drug B, we compare equipotent doses and not absolute doses (e.g. 0.1 mg of fentanyl is equivalent to 10 mg of morphine and when comparing potential benefits or side effects in certain cases of acute pain these doses are used and not 10 mg of fentanyl to 10 mg of morphine).To conclude, we have to say that we opted to perform each therapy as conduced in previous studies for the maximal duration of time recommended by the producer (added in the methods section). 
  2. In regard to the observation that the overall median treatment time differed between the groups (in terms of a median of 2 sessions in MARS group and 3 sessions in terms of CytoSorb group, not in term of absolute liver support therapy hours due to the reasons mentioned before) the observation made by the reviewer is correct. Moreover, the study was not powered for outcome measurement (survival, length of ICU stay etc.) and hence the main outcomes you would want for the entire treatment in comparison to a single session in either therapy could not be addressed. This is the reason we have focused our research on the benefits of a single session in improving laboratory parameters. We have added this in the discussion session in order to make this clearer for the reviewer. 

Reviewer 2 Report

Re: Artificial Liver Support with Cytosorb and MARS in Liver Failure: A Retrospective Propensity Matched Analysis

By: Mihai Popescu et al.

Thank you for the opportunity to review the above mentioned manuscript.

The authors present a study evaluating the two different methods (namely MARS and CytoSorb) for artificial liver support in critically ill patients with either acute or acute-on-chronic liver failure. In total n=30 patients were included, thereof n=15 in the MARS group and n=15 patients treated with CytoSorb.

The manuscript is clearly written and very informative. The topic is of relevance to clinicians working in the field of extracorporeal artificial liver support in patients with liver dysfunction.

The title is informative and reflects the aim of the study.

The introduction leads the reader into the topic and ends with aims of the study.

The methods are described clearly. With concerning to the statitics, the authors report on propensity score matching in the title of the manuscript. However, this technique and the matching process is not clearly reported in the statistic section and should be added.

The authors discuss the results of the study in a structured manner. Unfortunately, the discussion section is almost exclusively written in favor of CytoSorb. However, when comparing a single session, both systems performed equally and therefore the statement “..CytoSorb may provide a more extensive biochemical control of liver failure compared to MARS” (p10L311-312) is very questionable. Therefore, the discussion section should be reorganised.

While reading the manuscript some aspects appeared, which will be addressed below:

Major findings:

-        Patients were matched for age, severity scores, bilirubin, lactate and vasopressor rates. However, these parameters were not fully shown within the manuscript --- please add the missing variables.

-        MARS usually runs for a maximum duration of 8 hours (until adsorber columns were saturated). In contrast, Cytosorb is implemented for 24h into the dialysis circuit. The authors in the current manuscript report on similar duration of the therapies. Therefore, bias towards elimination and detoxification rates seem likely, as cytosorb runs 3 times longer. The authors could add reduction rates per hour to overcome this issue. Moreover, this issue needs to be discussed in the limitation section of the manuscript.

-        Results/figure1 and table 2: Reduction rates of bilirubin, LDH, ammonia and platelets within a single session were reported. I suggest reporting reduction rates (percentage) rather than reporting total values, to better compare both treatment modalities.

-        Section 3.1.2 Course of treatment, figure 2 and table 3: as I understand this section, results of all performed therapies in each of the patients were reported. – this is difficult in several perspectives: (1) patients in the cytosorb group were in median treated for 3 sessions, as patients in the MARS group received in median only 2 sessions. (2) as mentioned before, cytosorb was applied for 24h as MARS treatment only lasted for 8 hours. I seems therefore not surprising that cytosorb may have supported better than MARS treatment. – Please add a statement or correct.

-        ICU outcome: As 28d-survival was similar in both groups, the statement “….with a non-significant higher survival in the CytoSorb group…” seems therefore unnecessary, as there is no difference.

-        Please report indication thresholds for artificial liver support (i.e. MARS and CytoSorb) in your institution.

-        Discussion, p10L309-310: “Overall, …. Paraclinical improvement in liver functional tests demonstrated by …” --- what is meant by functional liver tests? – All presented values represent static liver parameter. Functional liver tests, i.e. plasma disappearance rate of ICG, were not reported - Please correct or describe.

Author Response

Dear Reviewer, 

Thank you very much for the time and effort to review our article. We have read them with great interest and we have made some substantial changes to the manuscript in order to make it more understandable to the reader. These have been marked with track changes throughout the manuscript. Below you can find a point by point discussion of the problems you have raised. 

Minor problems: 

  1. You are very right, we have omitted the description of the propensity matching. This has now been included in the statistical part of the methods.
  2. In respect to the discussion section that seems to be written in favor of CytoSorb you are right again. We have made the appropriate changes in this part of the article to make it as objective as possible. 

Response to major problems:

  1. you mentioned that "Patients were matched for age, severity scores, bilirubin, lactate and vasopressor rates. However, these parameters were not fully shown within the manuscript --- please add the missing variables." These variables were in the original manuscript in table 1.
  2. in regard to the difference in time of extracorporeal liver support therapy. the maximal time recommended by each of the companies who produce the respective filters as this is considered to be the time at which the filter is no more efficient (8 h for MARS and 24 h for CS). Moreover, this also is the time that has been considered the standard time for each of the procedure in the most important studies published in current literature: MARS (https://doi.org/10.1371/journal. pone.0175529, DOI: 10.1097/SLA.0000000000002361) including, of course, in the largest trial on the issue of using MARS in ACLF - the RELIEF trail (doi:10.1002/hep.26185) and for CS (doi:10.1177/0391398820981383, https://doi.org/10.3390/jcm10215182). If we are allowed to make a comparison, when comparing drug A to drug B, we compare equipotent doses and not absolute doses (e.g. 0.1 mg of fentanyl is equivalent to 10 mg of morphine and when comparing potential benefits or side effects in certain cases of acute pain these doses are used and not 10 mg of fentanyl to 10 mg of morphine). To conclude, we have to say that we opted to perform each therapy as conduced in previous studies for the maximal duration of time recommended by the producer (added in the methods section). 
  3. In regard to reporting the results as per hour. This would be valid if the two methods would have the same elimination kinetics. Due to the fact that MARS and CytoSorb has different elimination kinetics, and the fact that these kinetics are non-linear (as wonderfully demonstrated by Dominik et al in their in vivo study - DOI: 10.1159/000508810) reporting the results at "x" hours or even hourly would introduce an even bigger bias. Hence, as previously mentioned we have chosen to report the results in accordance with all published trials in the field at the end of the session (as this is the clinical important outcome) and has the lowest non-linear kinetics induced biased. 
  4. " Results/figure1 and table 2: Reduction rates of bilirubin, LDH, ammonia and platelets within a single session were reported. I suggest reporting reduction rates (percentage) rather than reporting total values, to better compare both treatment modalities". If the reviewer thinks this is a crucial issue (as it is included in the major comments) of course we will comply, but we consider that this is not necessary due to the following reasons: (a) the absolute numbers have the same relevance, adding both these numbers and % would make the tables/figures impossible to read due to high number of columns; (b) of this article were to be compared with others or be included in some meta-analysis the raw numbers are required and hence they would be unavailable; (c) as a liver intensivist when reading articles I am more interested in the absolute numbers as this allows me to compare "their" patients with my patients.
  5. "Section 3.1.2 Course of treatment, figure 2 and table 3: as I understand this section, results of all performed therapies in each of the patients were reported. – this is difficult in several perspectives: (1) patients in the cytosorb group were in median treated for 3 sessions, as patients in the MARS group received in median only 2 sessions. (2) as mentioned before, cytosorb was applied for 24h as MARS treatment only lasted for 8 hours. I seems therefore not surprising that cytosorb may have supported better than MARS treatment. – Please add a statement or correct." We think that difference in therapy duration has been clearly described above but the reviewer is perfectly correct about the difference in the number of sessions between the two groups and this may induce a significant bias. This is why we have introduced this in the discussion section as well as in the "limitations part".   
  6. "ICU outcome: As 28d-survival was similar in both groups, the statement “….with a non-significant higher survival in the CytoSorb group…” seems therefore unnecessary, as there is no difference." Your are right, we have changed this in the results section.   
  7. "Please report indication thresholds for artificial liver support (i.e. MARS and CytoSorb) in your institution." We have included this in the methods section. 
  8. Discussion, p10L309-310: “Overall, …. Paraclinical improvement in liver functional tests demonstrated by …” --- what is meant by functional liver tests? – All presented values represent static liver parameter. Functional liver tests, i.e. plasma disappearance rate of ICG, were not reported - Please correct or describe. We have corrected this in the discussions and only left "liver tests", of course that by "liver functional tests" we were thinking about tests looking at the liver function (synthesis, detoxification etc) that are routinely used in clinical practice. However, you are right and some readers may think about the dynamic liver functional tests and hence we have made the appropriate change to make things more cleared. 

Round 2

Reviewer 2 Report

Re: Artificial Liver Support with Cytosorb and MARS in Liver Failure: A Retrospective Propensity Matched Analysis

By: Mihai Popescu et al.

Thank you for the opportunity to review the revision of the above mentioned manuscript.

The authors present a  revised version of the inital manuscript after major revision. During the revision process the content of the manuscript improved and most of the suggested changes were applied.

Three minor findings still need to be clarified:

-          Table 2 is named twice -- Table 2. Comparison on clinical and paraclinical data and severity scores before and after a single session of liver support therapy and Table 2. Comparison on clinical and paraclinical data and severity scores before and after a liver support treatment. --- Please correct

-          Tables 2: what does the abbreviation “PCR” stands for (stands below PCT and above Firbrinogen)? –Please describe

-          Please check the whole manuscript for spelling errors and typos

Kind regards

Author Response

Dear Reviewer, 

Thank you once again for carefully reading our article and making the appropriate suggestions to improve the quality of our manuscript. 

In regard to the minor concerns raised: 

  1. Thank you for spotting this mistake. Indeed, both table 2 and 3 were titled table 2. We have corrected this and made the appropriate changes in the results section so that the text is correctly referenced to the appropriate table.
  2. In regard to "PCR" this was a typo in table 2. It should have stated CPR (the abbreviation of which is C-reactive protein as in the legend). We have made this change in both table 2 and table 3. 
  3. We have read throughout the article and corrected the spelling and grammar mistakes (including some upper scripts in platelet count) we have encountered.